# Enabling certification of verification-agnostic networks via memory-efficient semidefinite programming

**Sumanth Dathathri**[*1], **Krishnamurthy (Dj) Dvijotham**[*1], **Alex Kurakin**[*2],
**Aditi Raghunathan**[*3], **Jonathan Uesato**[*1], **Rudy Bunel**[1], **Shreya Shankar**[3],
**Jacob Steinhardt**[4], **Ian Goodfellow**[5], **Percy Liang**[3], **Pushmeet Kohli**[1]

[1]DeepMind [2]Google Brain [3]Stanford [4]UC Berkeley [5]Work done at Google

`{sdathath,dvij,kurakin,juesato}@google.com`, `aditir@stanford.edu`

## Abstract

Convex relaxations have emerged as a promising approach for verifying desirable properties of neural networks like robustness to adversarial perturbations. Widely used Linear Programming (LP) relaxations only work well when networks are trained to facilitate verification. This precludes applications that involve *verification-agnostic* networks, i.e., networks not specially trained for verification. On the other hand, semidefinite programming (SDP) relaxations have successfully be applied to verification-agnostic networks, but do not currently scale beyond small networks due to poor time and space asymptotics. In this work, we propose a first-order dual SDP algorithm that (1) requires memory only linear in the total number of network activations, (2) only requires a fixed number of forward/backward passes through the network per iteration. By exploiting iterative eigenvector methods, we express all solver operations in terms of forward and backward passes through the network, enabling efficient use of hardware like GPUs/TPUs. For two verification-agnostic networks on MNIST and CIFAR-10, we significantly improve $\ell_\infty$ verified robust accuracy from $1\% \rightarrow 88\%$ and $6\% \rightarrow 40\%$ respectively. We also demonstrate tight verification of a quadratic stability specification for the decoder of a variational autoencoder.

## 1   Introduction

Applications of neural networks to safety-critical domains requires ensuring that they behave as expected under all circumstances [32]. One way to achieve this is to ensure that neural networks conform with a list of *specifications*, i.e., relationships between the inputs and outputs of a neural network that ought to be satisfied. Specifications can come from safety constraints (a robot should never enter certain unsafe states [40, 29, 12]), prior knowledge (a learned physical dynamics model should be consistent with the laws of physics [49]), or stability considerations (certain transformations of the network inputs should not significantly change its outputs [57, 7]).

Evaluating whether a network satisfies a given specification is a challenging task, due to the difficulty of searching for violations over the high dimensional input spaces. Due to this, several techniques that claimed to enhance neural network robustness were later shown to break under stronger attacks [61, 5]. This has motivated the search for verification algorithms that can provide provable guarantees on neural networks satisfying input-output specifications.

Popular approaches based on linear programming (LP) relaxations of neural networks are computationally efficient and have enabled successful verification for many specifications [37, 18, 30, 21]. LP relaxations are sound (they would never incorrectly conclude that a specification is satisfied) but

incomplete (they may fail to verify a specification even if it is actually satisfied). Consequently, these approaches tend to give poor or vacuous results when used in isolation, though can achieve strong results when combined with specific training approaches to aid verification [22, 51, 67, 21, 54, 6].

In contrast, we focus on *verification-agnostic* models, which are trained in a manner agnostic to the verification algorithm. This would enable applying verification to all neural networks, and not just those trained to be verifiable. First, this means training procedures need not be constrained by the need to verify, thus allowing techniques which produce empirically robust networks, which may not be easily verified [38]. Second, ML training algorithms are often not easily modifiable, e.g. production-scale ML models with highly specific pipelines. Third, for many tasks, defining formal specifications is difficult, thus motivating the need to learn specifications from data. In particular, in recent work [24, 50, 66], natural perturbations to images like changes in lighting conditions or changes in the skin tone of a person, have been modeled using perturbations in the latent space of a generative model. In these cases, the specification itself is a verification-agnostic network which the verification must handle even if the prediction network is trained with the verification in mind.

In contrast to LP-based approaches, the semidefinite programming (SDP) relaxation [52] has enabled robustness certification of verification-agnostic networks. However, the interior point methods commonly used for SDP solving are computationally expensive with $O(n^6)$ runtime and $O(n^4)$ memory requirements, where $n$ is the number of neurons in the network [41, 60]. This limits applicability of SDPs to small fully connected neural networks.

Within the SDP literature, a natural approach is to turn to first-order methods, exchanging precision for scalability [63, 53]. Because verification only needs a bound on the optimal value of the relaxation (and not the optimal solution), we need not design a general-purpose SDP solver, and can instead operate directly in the dual. A key benefit is that the dual problem can be cast as minimizing the maximum eigenvalue of an affine function, subject only to non-negativity constraints. This is a standard technique used in the SDP literature [25, 42] and removes the need for an expensive projection operation onto the positive semidefinite cone. Further, since any set of feasible dual variables provides a valid upper bound, we do not need to solve the SDP to optimality as done previously [52], and can instead stop once a sufficiently tight upper bound is attained.

In this paper, we show that applying these ideas to neural network verification results in an efficient implementation both in theory and practice. Our solver requires $O(n)$ memory rather than $O(n^4)$ for interior point methods, and each iteration involves a constant number of forward and backward passes through the network.

**Our contributions.** The key contributions of our paper are as follows:

1. By adapting ideas from the first-order SDP literature [25, 42], we observe that the dual of the SDP formulation for neural network verification can be expressed as a maximum eigenvalue problem with only interval bound constraints. This formulation generalizes [52] without loss of tightness, and applies to any quadratically-constrained quadratic program (QCQP), including the standard adversarial robustness specification and a variety of network architectures.

   Crucially, when applied to neural networks, we show that subgradient computations are expressible purely in terms of forward or backward passes through layers of the neural network. Consequently, applying a subgradient algorithm to this formulation achieves per-iteration complexity comparable to a constant number of forward and backward passes through the neural network.

2. We demonstrate the applicability of first-order SDP techniques to neural network verification. We first evaluate our solver by verifying $\ell_\infty$ robustness of a variety of *verification-agnostic* networks on MNIST and CIFAR-10. We show that our approach can verify large networks beyond the scope of existing techniques. For these verification-agnostic networks, we obtain bounds an order of magnitude tighter than previous approaches (Figure 1). For an adversarially trained convolutional neural network (CNN) with no additional regularization on MNIST ($\epsilon = 0.1$), compared to LP relaxations, we improve the verified robust accuracy from $1\%$ to $88\%$. For the same training and architecture on CIFAR-10 ($\epsilon = 2/255$), the corresponding improvement is from $6\%$ to $40\%$ (Table 1).

3. To demonstrate the generality of our approach, we verify a different quadratic specification on the stability of the output of the decoder for a variational autoencoder (VAE). The upper bound on specification violation computed by our solver closely matches the lower bound on specification violation (from PGD attacks) across a wide range of inputs (Section 6.2).

## 2  Related work

**Neural network verification.** There is a large literature on verification methods for neural networks. Broadly, the literature can be grouped into complete verification using mixed-integer programming [26, 18, 59, 10, 2], bound propagation [56, 70, 65, 21], convex relaxation [30, 17, 67, 51], and randomized smoothing [35, 11]. Verified training approaches when combined with convex relaxations have led to promising results [30, 51, 23, 6]. Randomized smoothing and verified training approaches requires special modifications to the predictor (smoothing the predictions by adding noise) and/or the training algorithm (training with additional noise or regularizers) and hence are not applicable to the verification-agnostic setting. Bound propagation approaches have been shown to be special instances of LP relaxations [37]. Hence we focus on describing the convex relaxations and complete solvers, as the areas most closely related to this paper.

**Complete verification approaches.** These methods rely on exhaustive search to find counter-examples to the specification, using smart propagation or bounding methods to rule out parts of the search space that are determined to be free of counter-examples. The dominant paradigms in this space are Satisfiability Modulo Theory (SMT) [26, 18] and Mixed Integer Programming (MIP) [59, 10, 2]. The two main issues with these solvers are that: 1) They can take exponential time in the network size and 2) They typically cannot run on accelerators for deep learning (GPUs, TPUs).

**Convex relaxation based methods.** Much work has relied on linear programming (LP) or similar relaxations for neural-network verification [30, 17]. Bound propagation approaches can also be viewed as a special case of LP relaxations [37]. Recent work [54] put all these approaches on a uniform footing and demonstrated using extensive experiments that there are fundamental barriers in the tightness of these LP based relaxations and that obtaining tight verification procedures requires better relaxations. A similar argument in [52] demonstrated a large gap between LP and SDP relaxations even for networks with randomly chosen weights. Fazlyab et al. [19, 20] generalized the SDP relaxations to arbitrary network structures and activation functions. However, these papers use off-the-shelf interior point solvers to solve the resulting relaxations, preventing them from scaling to large CNNs. In this paper, we focus on SDP relaxations but develop customized solvers that can run on accelerators for deep learning (GPUs/TPUs) enabling their application to large CNNs.

**First-order SDP solvers.** While interior-point methods are theoretically compelling, the demands of large-scale SDPs motivate first-order solvers. Common themes within this literature include smoothing of nonsmooth objectives [42, 33, 14] and spectral bundle or proximal methods [25, 36, 45]. Conditional gradient methods use a sum of rank-one updates, and when combined with sketching techniques, can represent the primal solution variable using linear space [68, 69]. Many primal-dual algorithms [64, 63, 41, 4, 15] exploit computational advantages of operating in the dual – in fact, our approach to verification operates exclusively in the dual, thus sidestepping space and computational challenges associated with the primal matrix variable. Our formulation in Section 5.1 closely follows the eigenvalue optimization formulation from Section 3 of Helmberg and Rendl [25]. While in this work, we show that vanilla subgradient methods are sufficient to achieve practical performance for many problems, many ideas from the first-order SDP literature are promising candidates for future work, and could potentially allow faster or more reliable convergence. A full survey is beyond scope here, but we refer interested readers to Tu and Wang [60] and the related work of Yurtsever et al. [69] for excellent surveys.

## 3  Verification setup

**Notation.** For vectors $a$,$b$, we use $a \leqslant b$ and $a \geqslant b$ to represent element-wise inequalities. We use $\mathcal{B}_\epsilon(x)$ to denote the $\ell_\infty$ ball of size $\epsilon$ around input $x$. For symmetric matrices $X$,$Y$, we use $X \succeq Y$ to denote that $X - Y$ is positive semidefinite (i.e. $X - Y$ is a symmetric matrix with non-negative eigenvalues) We use $[x]^+$ to denote $\max(x,0)$ and $[x]^-$ for $\min(x,0)$. $\mathbf{1}$ represents a vector of all ones.

**Neural networks.** We are interested in verifying properties of neural network with $L$ hidden layers and $N$ neurons that takes input $x_0$. $x_i$ denotes the activations at layer $i$ and the concantenated vector $x = [x_0, x_1, x_2, \cdots, x_L]$ represents all the activations of the network. Let $\mathbb{L}_i$ denote an affine map corresponding to a forward pass through layer $i$, for e.g., linear, convolutional and average pooling layers. Let $\sigma_i$ is an element-wise activation function, for e.g., ReLU, sigmoid, tanh. In this work, we focus on feedforward networks where $x_{i+1} = \sigma_i\big(\mathbb{L}_i(x_i)\big)$.

**Verification.** We study verification problems that involve determining whether $\phi(x) \leqslant 0$ for network inputs $x_0$ satisfying $\ell_0 \leqslant x_0 \leqslant u_0$ where specification $\phi$ is a function of the network activations $x$.

$$\texttt{opt} =: \max_x \phi(x) \ \text{subject to} \ \underbrace{x_{i+1} = \sigma_i\big(\mathbb{L}_i(x_i)\big)}_{\text{Neural net constraints}}, \ \underbrace{\ell_0 \leqslant x_0 \leqslant u_0}_{\text{Input constraints}}. \tag{1}$$

The property is verified if $\texttt{opt} \leqslant 0$. In this work, we focus on $\phi$ which are quadratic functions. This includes several interesting properties like verification of adversarial robustness (where $\phi$ is linear), conservation of an energy in dynamical systems [49]), or stability of VAE decoders (Section 6.2). Note that while we assume $\ell_\infty$-norm input constraints for ease of presentation, our approach is applicable to any quadratic input constraint.

## 4 Lagrangian relaxation of QCQPs for verification

A starting point for our approach is the following observation from prior work—the neural network constraints in the verification problem (1) can be replaced with quadratic constraints for ReLUs [52] and other common activations [19], yielding a Quadratically Constrained Quadratic Program (QCQP). We bound the solution to the resulting QCQP via a Lagrangian relaxation. Following [52], we assume access to lower and upper bounds $\ell_i, u_i$ on activations $x_i$ such that $\ell_i \leqslant x_i \leqslant u_i$. They can be obtained via existing bound propagation techniques [65, 30, 70]. We use $\ell \leqslant x \leqslant u$ to denote the collection of activations and bounds at all the layers taken together.

We first describe the terms in the Lagrangian corresponding to the constraints encoding layer $i$ in a ReLU network: $x_{i+1} = \text{ReLU}(\mathbb{L}_i(x_i))$. Let $\ell_i, u_i$ denote the bounds such that $\ell_i \leqslant x_i \leqslant u_i$. We associate Lagrange multipliers $\lambda_i = [\lambda_i^{\text{a}}; \lambda_i^{\text{b}}; \lambda_i^{\text{c}}; \lambda_i^{\text{d}}]$ corresponding to each of the constraints as follows.

$$x_{i+1} \geqslant 0 \ [\lambda_i^{\text{a}}], \ x_{i+1} \geqslant \mathbb{L}_i(x_i) \ [\lambda_i^{\text{b}}]$$

$$x_{i+1} \odot \big(x_{i+1} - \mathbb{L}_i(x_i)\big) \leqslant 0 \ [\lambda_i^{\text{c}}], \ x_i \odot x_i - (\ell_i + u_i) \odot x_i + \ell_i \odot u_i \leqslant 0 \ [\lambda_i^{\text{d}}]. \tag{2}$$

The linear constraints imply that $x_{i+1}$ is greater than both $0$ and $\mathbb{L}_i(x_i)$. The first quadratic constraint together with the linear constraint makes $x_{i+1}$ equal to the larger of the two, i.e. $x_{i+1} = \max(\mathbb{L}_i(x_i), 0)$. The second quadratic constraint directly follows from the bounds on the activations. The Lagrangian $\mathcal{L}(x_i, x_{i+1}, \lambda_i)$ corresponding to the constraints and Lagrange multipliers described above is as follows.

$$\begin{aligned}
\mathcal{L}(x_i, x_{i+1}, \lambda_i) =\ & (-x_{i+1})^\top \lambda_i^{\text{a}} + (\mathbb{L}_i(x_i) - x_{i+1})^\top \lambda_i^{\text{b}} \\
& + \big(x_{i+1} \odot (x_{i+1} - \mathbb{L}_i(x_i))\big)^\top \lambda_i^{\text{c}} + (x_i \odot x_i - (\ell_i + u_i) \odot x_i + \ell_i \odot u_i)^\top \lambda_i^{\text{d}} \\
=\ & \underbrace{(\ell_i \odot u_i)^\top \lambda_i^{\text{d}}}_{\text{independent of } x_i, x_{i+1}} \underbrace{- x_{i+1}^\top \lambda_i^{\text{a}} + (\mathbb{L}_i(x_i))^\top \lambda_i^{\text{b}} - x_{i+1}^\top \lambda_i^{\text{b}} - x_i^\top \big((\ell_i + u_i) \odot \lambda_i^{\text{d}}\big)}_{\text{linear in } x_i, x_{i+1}} \\
& + \underbrace{x_{i+1}^\top \text{diag}(\lambda_i^{\text{c}}) x_{i+1} - x_{i+1}^\top \text{diag}(\lambda_i^{\text{c}}) \mathbb{L}_i(x_i) + x_i^\top \text{diag}(\lambda_i^{\text{d}}) x_i}_{\text{Quadratic in } x_i, x_{i+1}}. \tag{3}
\end{aligned}$$

The overall Lagrangian $\mathcal{L}(x, \lambda)$ is the sum of $\mathcal{L}(x_i, x_{i+1}, \lambda_i)$ across all layers together with the objective $\phi(x)$, and consists of terms that are either independent of $x$, linear in $x$ or quadratic in $x$. Thus, $\mathcal{L}(x, \lambda)$ is a quadratic polynomial in $x$ and can be written in the form $\mathcal{L}(x, \lambda) = c(\lambda) + x^\top g(\lambda) + \frac{1}{2} x^\top H(\lambda) x$. Each of the coefficients $c(\lambda)$, $g(\lambda)$, and $H(\lambda)$ are affine as a function of $\lambda$. We will describe our approach in terms of $c(\lambda)$, $g(\lambda)$, and $H(\lambda)$, which need not be derived by hand, and can instead be directly obtained from the Lagrangian $\mathcal{L}(x, \lambda)$ via automatic differentiation as we discuss in Section 5.2. We observe that $\mathcal{L}(x, \lambda)$ is itself composed entirely of forward passes $\mathbb{L}_i(x_i)$ and element-wise operations. This makes computing $\mathcal{L}(x, \lambda)$ both convenient to implement and efficient to compute in deep learning frameworks.

Via standard Lagrangian duality, the Lagrangian provides a bound on $\texttt{opt}$:

$$\texttt{opt} \leqslant \min_{\lambda \geqslant 0} \max_{\ell \leqslant x \leqslant u} \mathcal{L}(x, \lambda) = \min_{\lambda \geqslant 0} \max_{\ell \leqslant x \leqslant u} c(\lambda) + x^\top g(\lambda) + \frac{1}{2} x^\top H(\lambda) x. \tag{4}$$

We now describe our dual problem formulation starting from this Lagrangian (4).

## 5 Scalable and Efficient SDP-relaxation Solver

Our goal is to develop a custom solver for large-scale neural network verification with the following desiderata: (1) compute *anytime* upper bounds valid after each iteration, (2) rely on elementary

computations with efficient implementations that can exploit hardware like GPUs and TPUs, and (3) have per-iteration memory and computational cost that scales linearly in the number of neurons.

In order to satisfy these desiderata, we employ first order methods to solve the Langrange dual problem (4). We derive a reformulation of the Lagrange dual with only non-negativity constraints on the decision variables (Section 5.1). We then show how to efficiently and conveniently compute subgradients of the objective function in Section 5.2 and derive our final solver in Algorithm 1.

## 5.1 Reformulation to a problem with only non-negativity constraints

Several algorithms in the first-order SDP literature rely on reformulating the semidefinite programming problem as an eigenvalue minimization problem [25, 42]. Applying this idea, we obtain a Lagrange dual problem which only has non-negativity constraints and whose subgradients can be computed efficiently, enabling efficient projected subgradient methods to be applied.

Recall that $\ell_i, u_i$ denote precomputed lower and upper bounds on activations $x_i$. For simplicity in presentation, we assume $\ell_i = -\mathbf{1}$ and $u_i = \mathbf{1}$ respectively for all $i$. This is without loss of generality, since we can always center and rescale the activations based on precomputed bounds to obtain normalized activations $\bar{x} \in [-1,1]$ and express the Lagrangian in terms of the normalized activations $\bar{x}$.

*Proposition* 1. The optimal value $\mathtt{opt}$ of the verification problem (1) is bounded above by the Lagrange dual problem corresponding to the Lagrangian in (4) which can be written as follows:

$$\mathtt{opt}_{\text{relax}} =: \min_{\lambda \geqslant 0, \kappa \geqslant 0} \underbrace{c(\lambda) + \frac{1}{2}\mathbf{1}^\top \left[\kappa - \lambda_{\min}^-(\text{diag}(\kappa) - M(\lambda))\mathbf{1}\right]^+}_{f(\lambda,\kappa)}, \quad M(\lambda) = \begin{pmatrix} 0 & g(\lambda)^\top \\ g(\lambda) & H(\lambda) \end{pmatrix}, \quad (5)$$

and $\lambda_{\min}^-(Z) = \min(\lambda_{\min}(Z), 0)$ is the negative portion of the smallest eigenvalue of $Z$ and $\kappa \in \mathbb{R}^{1+N}$.

*Proof Sketch.* Instead of directly optimizing over the primal variables $x$ in the Lagrangian of the verification problem (4), we explicitly add the redundant constraint $x^2 \leqslant 1$ with associated dual variables $\kappa$, and then optimize over $x$ in closed form. This does not change the the primal (or dual) optimum, but makes the constraints in the dual problem simpler. In the corresponding Lagrange dual problem (now over $\lambda, \kappa$), there is a PSD constraint of the form $\text{diag}(\kappa) \succeq M(\lambda)$. Projecting onto this constraint directly is expensive and difficult. However, for any $(\lambda, \kappa) \geq 0$, we can construct a dual feasible solution $(\lambda, \hat{\kappa})$ by simply subtracting the smallest eigenvalue of $\text{diag}(\kappa) - M(\lambda)$, if negative. For any non-negative $\lambda, \kappa$, the final objective $f(\lambda, \kappa)$ is the objective of the corresponding dual feasible solution and the bound follows from standard Lagrangian duality. The full proof appears in Appendix A.3. □

*Remark 1.* Raghunathan et al. [52] present an SDP relaxation to the QCQP for the verification of $\ell_\infty$ adversarial robustness. The solution to their SDP is equal to $\mathtt{opt}_{\text{relax}}$ in our formulation (5) (Appendix A.4). Raghunathan et al. [52] solve the SDP via interior-point methods using off-the-shelf solvers which simply cannot scale to larger networks due to memory requirement that is quartic in the number of activations. In contrast, our algorithm (Algorithm 1) has memory requirements that scale linearly in the number of activations.

*Remark 2.* Our proof is similar to the standard maximum eigenvalue transformulation for the SDP dual, as used in Helmberg and Rendl [25] or Nesterov [42] (see Appendix A.6 for details). Crucially for scalable implementation, our formulation avoids explicitly computing or storing the matrices for either the primal or dual SDPs. Instead, we will rely on automatic differentiation of the Lagrangian and matrix-vector products to represent these matrices implicitly, and achieve linear memory and runtime requirements. We discuss this approach now.

## 5.2 Efficient computation of subgradients

Our formulation in (5) is amenable to first-order methods. Projections onto the feasible set are simple and we now show how to efficiently compute the subgradient of the objective $f(\lambda, \kappa)$. By Danskin's theorem [13],

$$\partial_{\lambda,\kappa}\left(c(\lambda) + \frac{1}{2}\left[\kappa - \left[v^{\star\top}\left(\text{diag}(\kappa) - M(\lambda)\right)v^\star\right]\mathbf{1}\right]^{+\top}\mathbf{1}\right) \in \partial_{\lambda,\kappa} f(\lambda,\kappa), \quad (6a)$$

$$\text{where } v^\star = \underset{\|v\|=1}{\text{argmin}}\, v^\top\left(\text{diag}(\kappa) - M(\lambda)\right)v = \text{eigmin}(\text{diag}(\kappa) - M(\lambda)), \quad (6b)$$

and $\partial_{\lambda,\kappa}$ denotes the subdiffirential with respect to $\lambda,\kappa$. In other words, given any eigenvector $v^\star$ corresponding to the minimum eigenvalue of the matrix $\mathrm{diag}(\kappa) - M(\lambda)$, we can obtain a valid subgradient by applying autodiff to the left-hand side of (6a) while treating $v^\star$ as fixed. [1] The main computational difficulty is computing $v^\star$. While our final certificate will use an exact eigendecomposition for $v^\star$, for our subgradient steps, we can approximate $v^\star$ using an iterative method such as Lanczos [34]. Lanczos only requires repeated applications of the linear map $\mathbb{A} =: v \mapsto \big(\mathrm{diag}(\kappa) - M(\lambda)\big)v$. This linear map can be easily represented via derivatives and Hessian-vector products of the Lagrangian.

**Implementing implicit matrix-vector products via autodiff.** Recall from Section 4 that the Lagrangian is expressible via forward passes through affine layers and element-wise operations involving adjacent network layers. Since $M(\lambda)$ is composed of the gradient and Hessian of the Lagrangian, we will show computing the map $M(\lambda)v$ is computationally roughly equal to a forwards+backwards pass through the network. Furthermore, implementing this map is extremely convenient in ML frameworks supporting autodiff like TensorFlow [1], PyTorch [47], or JAX [8]. From the Lagrangian (4), we note that

$$g(\lambda) = \mathcal{L}_x(0,\lambda) = \left.\frac{\partial\mathcal{L}(x,\lambda)}{\partial x}\right|_{0,\lambda} \quad \text{and} \quad H(\lambda)v = \mathcal{L}^v_{xx}(0,\lambda,v) = \left.\left(\frac{\partial^2\mathcal{L}(x,\lambda)}{\partial x \partial x^T}\right)\right|_{0,\lambda} v = \left.\frac{\partial v^\top \mathcal{L}_x(0,\lambda)}{\partial x}\right|_{0,\lambda}.$$

Thus, $g(\lambda)$ involves a single gradient, and by using a standard trick for Hessian-vector products [48], the Hessian-vector product $H(\lambda)v$ requires roughly double the cost of a standard forward-backwards pass, with linear memory overhead. From the definition of $M(\lambda)$ in (5), we can use the quantities above to get

$$\mathbb{A}[v] = \big(\mathrm{diag}(\kappa) - M(\lambda)\big)v = -\kappa \odot v + \begin{pmatrix} (g(\lambda))^\top v_{1:N} \\ g(\lambda)v_0 + H(\lambda)v_{1:N} \end{pmatrix} = \kappa \odot v - \begin{pmatrix} (\mathcal{L}_x(0,\lambda))^\top v_{1:N} \\ \mathcal{L}_x(0,\lambda)v_0 + \mathcal{L}^v_{xx}(0,\lambda,v_{1:N}) \end{pmatrix},$$

where $v_0$ is the first coordinate of $v$ and $v_{1:N}$ is the subvector of $v$ formed by remaining coordinates.

### 5.3 Practical tricks for faster convergence

The Lagrange dual problem is a convex optimization problem, and a projected subgradient method with appropriately decaying step-sizes converges to an optimal solution [43]. However, we can achieve faster convergence in practice through careful choices for initialization, regularization, and learning rates.

**Initialization.** Let $\kappa_{\mathrm{opt}}(\lambda)$ denote the value of $\kappa$ that optimizes the bound (5), for a fixed $\lambda$. We initialize with $\lambda = 0$, and the corresponding $\kappa_{\mathrm{opt}}(0)$ using the following proposition.

*Proposition* 2. For any choice of $\lambda$ satisfying $H(\lambda) = 0$, the optimal choice $\kappa_{\mathrm{opt}}(\lambda)$ is given by

$$\kappa_0^* = \sum_{i=1}^{n} |g(\lambda)|_i \quad ; \quad \kappa_{1:n}^* = |g(\lambda)|$$

where $\kappa = [\kappa_0; \kappa_{1:n}]$ is divided into a leading scalar $\kappa_0$ and vector $\kappa_{1:n}$, and $|g(\lambda)|$ is elementwise.

See Appendix A.7 for a proof. Note that when $\phi(x)$ is linear, $H(\lambda) = 0$ is equivalent to removing the quadratic constraints on the activations and retaining the linear constraints in the Lagrangian (4).

**Regularization.** Next, we note that there always exists an optimal dual solution satisfying $\kappa_{1:n} = 0$, because they are the Lagrange multipliers of a redundant constraint; full proof appears in Appendix A.5. However, $\kappa$ has an empirical benefit of smoothing the optimization by preventing negative eigenvalues of $\mathbb{A}$. This is mostly noticeable in the early optimization steps. Thus, we can regularize $\kappa$ through either an additional loss term $\sum\kappa_{1:n}$, or by fixing $\kappa_{1:n}$ to zero midway through optimization. In practice, we found that both options occasionally improve final performance.

**Learning rates.** Empirically, we observed that the optimization landscape varies significantly for dual variables associated with different constraints (such as linear vs. quadratic). In practice, we found that using adaptive optimizers [16] such as Adam [27] or RMSProp [58] was necessary to stabilize optimization. Additional learning rate adjustment for $\kappa_0$ and the dual variables corresponding to the quadratic ReLU constraints provided an improvement on some network architectures (see Appendix B).

### 5.4 Algorithm for verifying network specifications

**Algorithm 1** Verification via SDP-FO

**Input:** Specification $\phi$ and bounds on the inputs $\ell_0 \leqslant x_0 \leqslant u_0$
**Output:** Upper bound on the optimal value of (1)

**Bound computation**: Obtain layer-wise bounds $\ell, u = \text{BoundProp}(\ell_0, u_0)$ using approaches such as [39, 70]
**Lagrangian**: Define Lagrangian $\mathcal{L}(x, \lambda)$ from (4)
**Initialization**: Initialize $\lambda, \kappa$ (Section 5.3)
**for** $t = 1, ..., T$ **do**

    Define the linear operator $\mathbb{A}_t$ as $\mathbb{A}_t[v] = \kappa \odot v - \begin{pmatrix} (\mathcal{L}_x(0,\lambda))^\top v_{1:} \\ \mathcal{L}_x(0,\lambda)v_0 + \mathcal{L}_{xx}^v(0,\lambda,v_{1:}) \end{pmatrix}$ (see section 5.2)

    $v^\star \leftarrow \text{eigmin}(\mathbb{A}_t)$ using the Lanczos algorithm [34].

    Define the function $f_t(\lambda, \kappa) = \mathcal{L}(0, \lambda) + \left[ \kappa - \left[ v^{\star\top} \mathbb{A}_t[v^\star] \right] \mathbf{1} \right]^{+\top} \mathbf{1}$ (see (6))

    $\bar{f}_t \leftarrow f_t(\lambda^t, \kappa^t)$

    Update $\lambda^t, \kappa^t$ using any gradient based method to obtain $\tilde{\lambda}, \tilde{\kappa}$ with the gradients: $\frac{\partial}{\partial \lambda} f_t(\lambda^t, \kappa^t), \frac{\partial}{\partial \kappa} f_t(\lambda^t, \kappa^t)$

    Project $\lambda^{t+1} \leftarrow \left[ \tilde{\lambda} \right]^+, \kappa^{t+1} \leftarrow [\tilde{\kappa}]^+$.

**end for**
**return** $\min_t \bar{f}_t$

---

We refer to our algorithm (summarized in Algorithm 1) as SDP-FO since it relies on a first-order method to solve the SDP relaxation. Although the full algorithm involves several components, the implementation is simple (~100 lines for the core logic when implemented in JAX[9]) and easily applicable to general architectures and specifications. [2] SDP-FO uses memory linear in the total number of network activations, with per-iteration runtime linear in the cost of a forwards-backwards pass.

**Computing valid certificates.** Because Lanczos is an approximate method, we always report final bounds by computing $v^\star$ using a non-iterative exact eigen-decomposition method from SciPy [44]. In practice, the estimates from Lanczos are very close to the exact values, while using 0.2s/iteration on large convolutional network, compared to 5 minutes for exact eigendecomposition (see Appendix C).

# 6 Experiments

In this section, we evaluate our SDP-FO verification algorithm on two specifications: robustness to adversarial perturbations for image classifiers (Sec. 6.1), and robustness to latent space perturbations for a generative model (Sec. 6.2). In both cases, we focus on verification-agnostic networks.

## 6.1 Verification of adversarial robustness

**Metrics and baselines** We first study verification of $\ell_\infty$ robustness for networks trained on MNIST and CIFAR-10. For this specification, the objective $\phi(x)$ in (1) is given by $(x_L)_{y'} - (x_L)_y$, where $x_L$ denotes the the final network activations, i.e. logits, $y$ is the index of the true image label, and $y'$ is a target label. For each image and target label, we obtain a lower bound on the optimal $\underline{\phi}(x) \leqslant \phi^*(x)$ by running projected gradient descent (**PGD**) [38] on the objective $\phi(x)$ subject to $\ell_\infty$ input constraints. A verification technique provides upper bounds $\overline{\phi}(x) \geqslant \phi^*(x)$. An example is said to be verified when the worst-case upper bound across all possible labels, denoted $\overline{\phi}_x$, is below 0. We first compare (**SDP-FO, Algorithm 1** to the **LP** relaxation from [18], as this is a widely used approach for verifying large networks, and is shown by [55] to encompass other relaxations including [30, 17, 65, 70, 39, 23]. We further compare to the SDP relaxation from [51] solved using MOSEK [3], a commercial interior point SDP (**SDP-IP**) solver, and the **MIP** approach from [59].

**Models** Our main experiments on CNNs use two architectures: **CNN-A** from [67] and **CNN-B** from [6]. These contain roughly 200K parameters + 10K activations, and 2M parameters + 20K activations, respectively. All the networks we study are verification-agnostic: trained only with nominal and/or adversarial training [38], without any regularization to promote verifiability.

While these networks are much smaller than modern deep neural networks, they are an order of magnitude larger than previously possible for verification-agnostic networks. To compare with prior

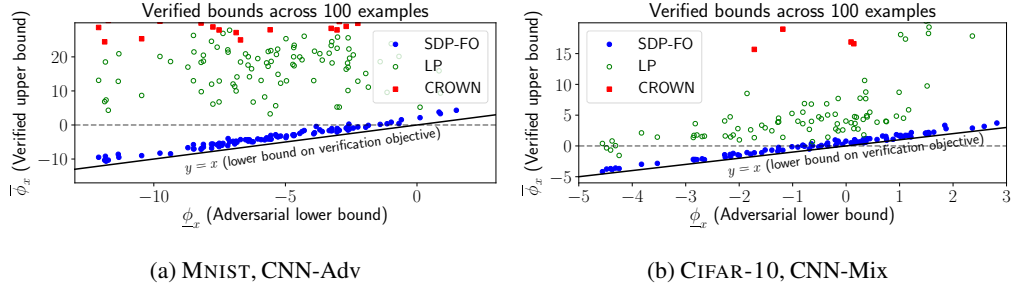

|  (a) MNIST, CNN-Adv | (b) CIFAR-10, CNN-Mix |

Figure 1: *Enabling certification of verification-agnostic networks.* For 100 random examples on MNIST and CIFAR-10, we plot the verified upper bound on $\phi_x$ against the adversarial lower bound (taking the worst-case over target labels for each). Recall, an example is verified when the verified upper bound $\overline{\phi}_x < 0$. Our key result is that SDP-FO achieves tight verification across all examples, with all points lying close to the line $y = x$. In contrast, LP or CROWN bounds produce much looser gaps between the lower and upper bounds. We note that many CROWN bounds exceed the plotted y-axis limits.

work, we also evaluate a variety of fully-connected **MLP** networks, using trained parameters from [51, 55]. These each contain roughly 1K activations. Complete training and hyperparameter details are included in Appendix B.1.

**Scalable verification of verification-agnostic networks** Our central result is that, for verification-agnostic networks, SDP-FO allows us to tractably provide significantly stronger robustness guarantees in comparison with existing approaches. In Figure 1, we show that SDP-FO reliably achieves tight verification, despite using loose initial lower and upper bounds obtained from CROWN [70] in Algorithm 1. Table 1 summarizes results. On all networks we study, we significantly improve on the baseline verified accuracies. For example, we improve verified robustness accuracy for CNN-A-Adv on MNIST from 0.4% to 87.8% and for CNN-A-Mix on CIFAR-10 from 5.8% to 39.6%.

| Dataset | Epsilon | Model | Accuracy | | Verified Accuracy | | | |
| | | | Nominal | PGD | SDP-FO (Ours) | SDP-IP[†] | LP | MIP[†] |
|---|---|---|---|---|---|---|---|---|
| MNIST | $\epsilon = 0.1$ | MLP-SDP [52] | 97.6% | 86.4% | **85.2%** | 80% | 39.5% | 69.2% |
| | | MLP-LP [52] | 92.8% | 81.2% | **80.2%** | 80% | 79.4% | - |
| | | MLP-Adv [52] | 98.4% | 93.4% | **91.0%** | 82% | 26.6% | - |
| | | MLP-Adv-B [55] | 96.8% | 84.0% | **79.2%** | - | 33.2% | 34.4% |
| | | CNN-A-Adv | 99.1% | 95.2% | **87.8%** | - | 0.4% | - |
| | $\epsilon = 0.05$ | MLP-Nor [55] | 98.0% | 46.6% | **28.0%** | - | 1.8% | 6.0% |
| CIFAR-10 | $\epsilon = \frac{2}{255}$ | CNN-A-Mix-4 | 67.8% | 55.6% | **47.8%** | * | 26.8% | - |
| | | CNN-B-Adv-4 | 72.0% | 62.0% | **46.0%** | * | 20.4% | - |
| | | CNN-A-Mix | 74.2% | 53.0% | **39.6%** | * | 5.8% | - |
| | | CNN-B-Adv | 80.3% | 64.0% | **32.8%** | * | 2.2% | - |

[†] Using numbers from [52] for SDP-IP and [54] using approach of [59] for MIP. Dashes indicate previously reported numbers are unavailable.
* Computationally infeasible due to quartic memory requirement.

Table 1: Comparison of verified accuracy across verification algorithms. Highlighted rows indicate models trained in a verification-agnostic manner. All numbers computed across the same 500 test set examples, except when using previously reported values. For all networks, SDP-FO outperforms previous approaches. The improvement is largest for verification-agnostic models.

**Comparisons on small-scale problems** We empirically compare SDP-FO against SDP-IP using MOSEK, a commercial interior-point solver. Since the two formulations are equivalent (see Appendix A.4), solving them to optimality should result in the same objective. This lets us carefully isolate the effectiveness of the optimization procedure relative to the SDP relaxation gap. However, we note that for interior-point methods, the memory requirements are quadratic in the size of $M(\lambda)$, which becomes quickly intractable e.g. $\approx 10$ petabytes for a network with 10K activations. This restricts our comparison to the small MLP networks from [52], while SDP-FO can scale to significantly larger networks.

In Figure 4 of Appendix C.1, we confirm that on a small random subset of matching verification instances, SDP-FO bounds are only slightly worse than SDP-IP bounds. This suggests that optimization is typically not an issue for SDP-FO, and the main challenge is instead tightening the SDP relaxation. Indeed, we can tighten the relaxation by using CROWN precomputed bounds [70] rather than interval

arithmetic bounds [39, 22], which almost entirely closes the gap between SDP-FO and PGD for the first three rows of Table 1, including the verification-agnostic MLP-Adv. Finally, compared to numbers reported in [55], SDP-FO outperforms the MIP approach using progressive LP bound tightening [59].

**Computational resources** We cap the number of projected gradient iterations for SDP-FO. Using a P100 GPU, maximum runtime is roughly 15 minutes per MLP instances, and 3 hours per CNN instances, though most instances are verified sooner. For reference, SDP-IP uses 25 minutes on a 4-core CPU per MLP instance [52], and is intractable for CNN instances due to quartic memory usage.

**Limitations.** In principle, our solver's linear asymptotics allow scaling to extremely large networks. However, in practice, we observe loose bounds with large networks. In Table 1, there is already a significantly larger gap between the PGD and SDP-FO bounds for the larger CNN-B models compared to their CNN-A counterparts, and in preliminary experiments, this gap increases further with network size. Thus, while our results demonstrate that the SDP relaxation remains tight on significantly larger networks than those studied in Raghunathan et al. [52], additional innovations in either the formulation or optimization process are necessary to enable further scaling.

## 6.2 Verifying variational auto-encoders (VAEs)

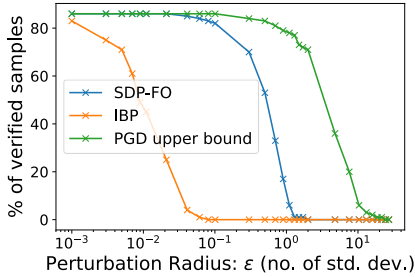

Figure 2: Comparison of different approaches for verifying the robustness of the decoder of a VAE on MNIST, measured across 100 samples. The lower-bound on the robust accuracy computed with SDP-FO closely matches the upper bound based on a PGD adversarial attack upto perturbations of 0.1 $\sigma_z$, while the lower bound based on IBP begins to diverge from the PGD upper bound at much smaller perturbations.

**Setup** To test the generality of our approach, we consider a different specification of verifying the validity of constructions from deep generative models, specifically variational auto-encoders (VAEs) [28]. Let $q_E(z|s) = \mathcal{N}(\mu_E^{z;s}, \sigma_E^{z;s})$ denote the distribution of the latent representation $z$ corresponding to input $s$, and let $q_D(s|z) = \mathcal{N}(\mu_D^{s;z}, I)$ denote the decoder. Our aim is to certify robustness of the decoder to perturbations in the VAE latent space. Formally, the VAE decoder is robust to $\ell_\infty$ latent perturbations for input $s$ and perturbation radius $\alpha \in \mathbb{R}^{++}$ if:

$$\varepsilon_{\text{recon}}(s, \mu_D^{s;z}) := \|s - \mu_D^{s;z}\|_2^2 \leq \tau \qquad \forall z' \text{ s.t } \|z' - \mu_E^{z;s}\|_\infty \leq \alpha \sigma_E^{z;s}, \qquad (7)$$

where $\varepsilon_{\text{recon}}$ is the reconstruction error. Note that unlike the adversarial robustness setting where the objective was linear, the objective function $\varepsilon_{\text{recon}}$ is quadratic. Quadratic objectives are not directly amenable to LP or MIP solvers without further relaxing the quadratic objective to a linear one. For varying perturbation radii $\alpha$, we measure the test set fraction with verified reconstruction error below $\tau = 40.97$, which is the median squared Euclidean distance between a point $s$ and the closest point with a different label (over MNIST).

**Results** We verify a VAE on MNIST with a convolutional decoder containing $\approx$ 10K total activations. Figure 2 shows the results. To visualize the improvements resulting from our solver, we include a comparison with guarantees based on interval arithmetic bound propagation (IBP) [23, 39], which we use to generate the bounds used in Algorithm 1. Compared to IBP, SDP-FO can successfully verify at perturbation radii roughly 50x as large. For example, IBP successfully verifies 50% at roughly $\epsilon = 0.01$ compared to $\epsilon = 0.5$ for SDP-FO. We note that besides the IBP bounds being themselves loose compared to the SDP relaxations, they further suffer from a similar drawback as LP/MIP methods in that they bound $\varepsilon_{\text{recon}}$ via $\ell_\infty$-bounds, which further results in looser bounds on $\varepsilon_{\text{recon}}$. Further details and visualizations are included in Appendix B.2.

## 7 Conclusion

We have developed a promising approach to scalable tight verification and demonstrated good performance on larger scale than was possible previously. While in principle, this solver is applicable to arbitrarily large networks, further innovations (in either the formulation or solving process) are necessary to get meaningful verified guarantees on larger networks.

**Acknowledgements**

We are grateful to Yair Carmon, Ollie Hinder, M Pawan Kumar, Christian Tjandraatmadja, Vincent Tjeng, and Rahul Trivedi for helpful discussions and suggestions. This work was supported by NSF Award Grant no. 1805310. AR was supported by a Google PhD Fellowship and Open Philanthropy Project AI Fellowship.

## Broader Impact

Our work enables verifying properties of verification-agnostic neural networks trained using procedures agnostic to any specification verification algorithm. While the present scalability of the algorithm does not allow it to be applied to SOTA deep learning models, in many applications it is vital to verify properties of smaller models running safety-critical systems (learned controllers running on embedded systems, for example). The work we have presented here does not address data related issues directly, and would be susceptible to any biases inherent in the data that the model was trained on. However, as a verification technique, it does not enhance biases present in any pre-trained model, and is only used as a post-hoc check. We do not envisage any significant harmful applications of our work, although it may be possible for adversarial actors to use this approach to verify properties of models designed to induce harm (for example, learning based bots designed to break spam filters or induce harmful behavior in a conversational AI system).

## Footnotes

\* Equal contribution. Alphabetical order.

† Code available at `https://github.com/deepmind/jax_verify`.

[1] The subgradient is a singleton except when the multiplicity of the minimum eigenvalue is greater than one, in which case any minimal eigenvector yields a valid subgradient.

[2] Core solver implementation at https://github.com/deepmind/jax_verify/blob/master/src/sdp_verify/sdp_verify.py

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
