[Supplementary Material]

# A Omitted proofs

## A.1 Adversarial robustness as quadratic specification

Consider certifying robustness: For input $x_0 \in \mathbb{R}^d$ with true label $i$, the network does not misclassify any adversarial example within $\ell_\infty$ distance of $\epsilon$ from $x_0$. This property holds if the score of any incorrect class $j$ is always lower than that of $i$ for all perturbations. Thus $\phi(x) = c^\top x_L$ with $c_j = 1$ and $c_i = -1$. The input constraints are also linear: $-\epsilon \leqslant x_i - x_{0i} \leqslant \epsilon$, for $i = 1,2,...d$.

## A.2 Linear and quadratic constraints for ReLU networks

**ReLU as quadratic constraints:** For the case of ReLU networks, we can do this exactly, as described in [52]. Consider a single activation $x_{\text{post}} = \max(x_{\text{pre}}, 0)$. This can be equivalently written as $x_{\text{post}} \geqslant 0, x_{\text{post}} \geqslant x_{\text{pre}}$, stating that $x_{\text{post}}$ is greater than $0$ and $x_{\text{pre}}$. Additionally, the quadratic constraint $x_{\text{post}}(x_{\text{post}} - x_{\text{pre}}) = 0$, enforces that $x_{\text{post}}$ is atleast one of the two. This can be extended to all units in the network allowing us to replace ReLU constraints with quadratic constraints.

## A.3 Formulation of bound constrained dual problem

*Proposition* 1. The optimal value $\texttt{opt}$ of the quadratic verification problem (1) is bounded above by

$$\texttt{opt}_{\text{relax}} =: \min_{\lambda \geqslant 0, \kappa \geqslant 0} \underbrace{c(\lambda) + \frac{1}{2}\mathbf{1}^\top \left[\kappa - \lambda_{\min}^-(\text{diag}(\kappa) - M(\lambda))\mathbf{1}\right]^+}_{f(\lambda,\kappa)}, \quad M(\lambda) = \begin{pmatrix} 0 & g(\lambda)^\top \\ g(\lambda) & H(\lambda) \end{pmatrix}, \quad (8)$$

and $\lambda_{\min}^-(Z)$ is the negative portion of the smallest eigen value of $Z$, i.e. $[\lambda_{\min}(Z)]^-$ and $\kappa \in \mathbb{R}^{1+N}$.

*Proof.* We start with the Lagrangian in (4) with rescaled activations such that $\ell = -\mathbf{1}$ and $u = \mathbf{1}$, where $\ell$ and $u$ are lower and upper bounds on the activations $x \in \mathbb{R}^n$ respectively. This normalization is achieved by using pre-computed bounds via bound propagation, which are used to write the quadratic constraints, as in [52].

$$\texttt{opt} \leqslant \min_{\lambda \geq 0} \max_{-1 \leq x \leq 1} \left(c(\lambda) + g(\lambda)^\top x + \frac{1}{2}x^\top H(\lambda)x\right). \quad (9)$$

Define

$$\tilde{X} = \begin{pmatrix} 1 & x^\top \\ x & xx^\top. \end{pmatrix}$$

In terms of the above matrix, the above Lagrangian relaxation (9) can be equivalently written as:

$$\texttt{opt} \leqslant =: \min_{\lambda \geq 0} \max_{\text{diag}(\tilde{X}) \leqslant 1} c(\lambda) + \frac{1}{2}\langle M(\lambda), \tilde{X} \rangle, \text{ where} \quad (10)$$

$$M(\lambda) = \begin{pmatrix} 0 & g(\lambda)^\top \\ g(\lambda) & H(\lambda) \end{pmatrix} \quad (11)$$

Note that $\tilde{X}$ is always a PSD matrix with diagonal entries bounded above by 1. This yields the following relaxation of (9)

$$\texttt{opt} \leqslant \texttt{opt}_{\text{sdp}} =: \min_{\lambda \geq 0} \max_{\text{diag}(X) \leq 1, X \geq 0} c(\lambda) + \frac{1}{2}\langle M(\lambda), X \rangle. \quad (12)$$

We introduce a Lagrange multiplier $\frac{1}{2}\kappa \in \mathbb{R}^{n+1}$ for the constraint $\text{diag}(X) \leq 1$[3]. Since $\texttt{opt}_{\text{sdp}}$ is convex, by strong duality, we have

$$\texttt{opt}_{\text{sdp}} = \min_{\lambda, \kappa \geq 0} \max_{X \geq 0} c(\lambda) + \frac{1}{2}\left(\langle M(\lambda), X \rangle + \kappa^\top \mathbf{1} - \langle \text{diag}(\kappa), X \rangle\right), \quad (13)$$

$$= \min_{\lambda, \kappa \geq 0} c(\lambda) + \frac{1}{2}\kappa^\top \mathbf{1} \text{ s.t. } \text{diag}(\kappa) - M(\lambda) \geq 0, \quad (14)$$

where the last equality follows from the facts that (i) when $\mathrm{diag}(\kappa) - M(\lambda)$ is not PSD, $\langle M(\lambda) - \mathrm{diag}(\kappa), X \rangle$ would be unbounded when maximizing over PSD matrices $X$ and (ii) when $\mathrm{diag}(\kappa) - M(\lambda) \geq 0$, the maximum value of inner maximization over PSD matrices $X$ is $0$.

Projecting onto the PSD constraint $\mathrm{diag}(\kappa) - M(\lambda) \geq 0$ directly is still expensive. Instead, we take the following approach. For any non-negative $(\kappa, \lambda)$, we generate a *feasible* $(\hat{\kappa}, \hat{\lambda})$ as follows.

$$\hat{\kappa} = \left[ \kappa - \lambda_{\min}^{-} \left[ \mathrm{diag}(\kappa) - M(\lambda) \right] \mathbf{1} \right]^{+}, \ \hat{\lambda} = \lambda \tag{15}$$

In other words, $\lambda$ remains unchanged with $\hat{\lambda} = \lambda$. To obtain $\hat{\kappa}$, we first compute the minimum eigenvalue $\lambda_{\min}$ of the matrix $\mathrm{diag}(\kappa) - M(\lambda)$. If this is positive, $(\kappa, \lambda)$ are feasible, and $\hat{\kappa} = \kappa, \hat{\lambda} = \lambda$. However, if this is negative, we then add the negative portion of $\lambda_{\min}^{-} = [\lambda_{\min}]^{-}$ to the diagonal matrix to make $\mathrm{diag}(\hat{\kappa}) \geq M(\lambda)$, and subsequently project onto the non-negativity constraint. The subsequent projection never decreases the value of $\hat{\kappa}$ and hence $\mathrm{diag}(\hat{\kappa}) - M(\lambda) \geq 0$.

Plugging $\hat{\kappa}, \hat{\lambda}$ in the objective above, and removing the PSD constraint gives us the following final formulation.

$$\mathtt{opt}_{\mathrm{sdp}} = \mathtt{opt}_{\mathrm{relax}} =: \min_{\lambda, \kappa \geq 0} c(\lambda) + \frac{1}{2} \left[ \kappa - \lambda_{\min}^{-}(\mathrm{diag}(\kappa) - M(\lambda)) \mathbf{1} \right]^{+\top} \mathbf{1}. \tag{16}$$

Note that feasible $\kappa, \lambda$ remain unchanged and hence the equality. $\qquad\square$

## A.4 Relaxation comparison to Raghunathan et al. [52]

Our solver (Algorithm 1) uses the formulation described in (5), replicated above in (16). In this section, we show that the above formulation is equivalent to the SDP formulation in [52] when we use quadratic constraints to replace the ReLU constraints, as done in [52] and presented above in Appendix A.2. We show this by showing equivalence with an intermediate SDP formulation below. From Appendix A.3, the solution to this intermediate fomulation matches that of relaxation we optimize (16).

$$\mathtt{opt}_{\mathrm{sdp}} =: \min_{\lambda \geq 0} \max_{\mathrm{diag}(X) \leqslant 1, X \geq 0} c(\lambda) + \frac{1}{2} \langle M(\lambda), X \rangle. \tag{17}$$

To mirror the block structure in $M(\lambda)$, we write $X \geq 0$ as follows.

$$X = \begin{pmatrix} X_{11} & X_x^\top \\ X_x & X_{xx} \end{pmatrix}, X_{xx} \geq \frac{1}{X_{11}} X_x X_x^\top, \tag{18}$$

where the last condition follows by Schur complements.

The objective then takes the form $\max\limits_{\mathrm{diag}(X_{xx}) \leq 1, X_{11} \leqslant 1} g(\lambda)^\top X_x + \frac{1}{2} \langle H(\lambda), X_{xx} \rangle$. Note that the feasible set (over $X_{xx}, X_x$) for $X_{11} = 1$ contains the feasible sets for any smaller $X_{11}$, by the Schur complement condition above. Since $X_{11}$ does not appear in the objective, we can set $X_{11} = 1$ to obtain the following equality.

$$\mathtt{opt}_{\mathrm{sdp}} = \min_{\lambda \geq 0} \max_{\mathrm{diag}(X) \leq 1, X_{11} = 1, X \geq 0} c(\lambda) + g(\lambda)^\top X_x + \frac{1}{2} \langle H(\lambda), X_{xx} \rangle, \tag{19}$$

where $X_{11}$ is the first entry, and $X_x, X_{xx}$ are the blocks as described in (18).

**Prior SDP.** Now we start with the SDP formulation in [52]. Recall that we have a QCQP that represents the original verification problem with quadratic constraints on activations. The relaxation in [52] involves intoducing a new matrix variable $P$ as follows.

$$P = \begin{pmatrix} P[1] & P[x] \\ P[x] & P[xx]. \end{pmatrix} \tag{20}$$

The quadratic constraints are now written in terms of $P$ where $P[x]$ replaces the linear terms and $P[xx]$ replaces the quadratic terms. Raghunathan et al. [52] optimize this primal SDP formulation to obtain $\mathtt{opt}_{\mathrm{prior\text{-}sdp}}$. By strong duality, $\mathtt{opt}_{\mathrm{prior\text{-}sdp}}$ matches the optimum of the dual problem obtained

via the Lagrangian relaxation of the SDP. In terms of the quantities $g, H$ that we defined in this work ((3) and (4)), we have

$$\texttt{opt}_{\text{prior-sdp}} = \min_{\lambda \geq 0} \max_{\text{diag}(P) \leq 1, P[1]=1, P \succeq 0} \mathcal{L}_{\text{prior-sdp}}(P, \lambda) \tag{21}$$

$$= \min_{\lambda \geq 0} \max_{\text{diag}(P) \leq 1, P[1]=1, P \succeq 0} c(\lambda) + g(\lambda)^\top P[x] + \frac{1}{2} \langle H(\lambda), P[xx] \rangle. \tag{22}$$

By redefining matrix $P$ as $X$, from (19) and (21), we have $\texttt{opt}_{\text{sdp}} = \texttt{opt}_{\text{prior-sdp}}$. From (16), we have $\texttt{opt}_{\text{sdp}} = \texttt{opt}_{\text{relax}}$ and hence proved that the optimal solution of our formulation matches that of prior work [52] when using the same quadratic constraints as used in [52]. In other words, our reformulation that allows for a subgradient based memory efficient solver does *not* introduce additional looseness over the original formulation that uses a memory inefficient interior point solver.

## A.5   Regularization of $\kappa$ via alternate dual formulation

In Section 5.3, we describe that it can be helpful to regularize $\kappa_{1:n}$ towards 0. This is motivated by the following proposition:

*Proposition* 3. The optimal value $\texttt{opt}$ is upper-bounded by the alternate dual problem

$$\texttt{opt} \leq \min_{\lambda, \kappa \geq 0} \underbrace{c(\lambda) + \frac{1}{2} \kappa_0}_{\hat{f}(\lambda, \kappa_0)} \text{ s.t. } \begin{pmatrix} \kappa_0 & -g(\lambda)^\top \\ -g(\lambda) & -H(\lambda) \end{pmatrix} \succeq 0 \tag{23}$$

Further, for any feasible solution $\lambda, \kappa_0$ for this dual problem, we can obtain a corresponding solution to $\texttt{opt}_{\text{relax}}$ with $\lambda, \kappa_0, \kappa_{1:n} = 0, \kappa = (\kappa_0; \kappa_{1:n})$, such that $f(\lambda, \kappa) = \hat{f}(\lambda, \kappa_0)$.

*Proof.* We begin with the Lagrangian dual

$$\texttt{opt} \leq \texttt{opt}_{\text{lagAlt}} =: \min_{\lambda \geq 0} \max_x c(\lambda) + x^\top g(\lambda) + \frac{1}{2} x^\top H(\lambda) x. \tag{24}$$

Note that this is exactly the dual from Equation (4), without the bound constraints on $x$ in the inner maximization. In other words, whereas Equation (4) encodes the bound constraints into both the Lagrangian and the inner maximization constraints, in Equation 24, the bound constraints are encoded in the Lagrangian only.

The inner maximization can be solved in closed form, and is maximized for $x = -H(\lambda)^{-1} g(\lambda)$, yielding

$$\texttt{opt}_{\text{lagAlt}} = \min_{\lambda \geq 0} c(\lambda) - \frac{1}{2} g(\lambda)^\top H(\lambda) g(\lambda). \tag{25}$$

We can then reformulate using Schur complements:

$$\texttt{opt}_{\text{lagAlt}} = \min_{\lambda \geq 0, \kappa_0} c(\lambda) + \frac{1}{2} \kappa_0 \text{ s.t. } \kappa_0 \geq -g(\lambda) H(\lambda)^{-1} g(\lambda) \tag{26a}$$

$$= \min_{\lambda \geq 0, \kappa_0} c(\lambda) + \frac{1}{2} \kappa_0 \text{ s.t. } \hat{M}(\lambda) \succeq 0 \text{ where} \tag{26b}$$

$$\hat{M}(\lambda) = \begin{pmatrix} \kappa_0 & -g(\lambda)^\top \\ -g(\lambda) & -H(\lambda) \end{pmatrix}. \tag{26c}$$

To see that this provides a corresponding solution to $\texttt{opt}_{\text{relax}}$, we note that when $\hat{M} \succeq 0$, the choice $\kappa = (\kappa_0; \kappa_{1:n}), \kappa_{1:n} = 0$ makes $\text{diag}(\kappa) - M(\lambda) = \hat{M}(\lambda)$, and so $\lambda_{\min}^- [\text{diag}(\kappa) - M(\lambda)] = 0$. Thus, for any solution $\lambda, \kappa_0$, we have $f(\lambda, \kappa) = \hat{f}(\lambda, \kappa_0) = c(\lambda) + \frac{1}{2} \kappa_0$.

$\square$

*Remark.* Proposition 3 indicates that regularizing $\kappa_{1:n}$ towards 0 corresponds to solving the alternate dual formulation $\texttt{opt}_{\text{dualAlt}}$, which does not use bound constraints for the inner maximization. In this case, the role of $\kappa_{1:n}$ and $\hat{\kappa}$ is slightly different: even in the case when $\kappa_{1:n}$ is clamped to 0, the bound-constrained formulation allows an efficient projection operator, which in turn provide efficient any-time bounds.

## A.6 Informal comparison to standard maximum eigenvalue formulation

Our derivation for Proposition 1 is similar to maximum eigenvalue formulations for dual SDPs – our main emphasis is that when applied to neural networks, we can use autodiff and implicit matrix-vector products to efficiently compute subgradients.

We also mention here a minor difference in derivations for convenience of readers. The common derivation for these maximum eigenvalue formulations starts with an SDP primal under the assumption that all feasible solutions for the matrix variable $X$ have fixed trace. This trace assumption plays an analogous role to our interval constraints in the QCQP (12). These interval constraints also imply a trace constraint (since $\mathrm{diag}(X) \leqslant 1$ implies $\mathrm{tr}(X) \leqslant N+1$), but the interval constraints also allow us to use $\kappa$ to smooth the optimization. Without $\kappa$, any positive eigenvalues of $M(\lambda)$ cause large spikes in the objective – simplifying the objective $f(\lambda, \kappa)$ in (5) reveals the term $(N+1)\lambda_{\max}^{+}(M(\lambda))$ which grows linearly with $N$. As expected, this term also appears in these other formulations [25, 42].

## A.7 Proof of Proposition 2

*Proposition* 2. For any choice of $\lambda$ satisfying $H(\lambda) = 0$, the optimal choice $\kappa_{\mathrm{opt}}(\lambda)$ is given by

$$\kappa_0^* = \sum_{i=1}^{n} |g(\lambda)|_i \quad ; \quad \kappa_{1:n}^* = |g(\lambda)|$$

where we have divided $\kappa = [\kappa_0; \kappa_{1:n}]$ into a leading scalar $\kappa_0$ and a vector $\kappa_{1:n}$.

*Proof.* We use the dual expression from Equation (14):

$$\mathtt{opt}_{\mathrm{sdp}} = \min_{\lambda, \kappa \geq 0} \; c(\lambda) + \frac{1}{2}\kappa^\top \mathbf{1} \text{ s.t. } \mathrm{diag}(\kappa) - M(\lambda) \succeq 0.$$

Notice that by splitting $\kappa$ into its leading component $\kappa_0$ (a scalar) and the subvector $\kappa_{1:n} = [\kappa_1, ..., \kappa_n]$ (a vector of the same dimension as $x$), the constraint between $\kappa, \lambda$ evaluates to

$$\mathrm{diag}(\kappa) - M(\lambda) = \begin{pmatrix} \kappa_0 & g(\lambda)^\top \\ g(\lambda) & \mathrm{diag}(\kappa_{1:n}) \end{pmatrix} \succeq 0$$

Using Schur complements, we can rewrite the PSD constraint as

$$\Big( \mathrm{diag}(\kappa) - M(\lambda) \Big) \succeq 0 \Leftrightarrow \kappa_0 \geqslant \sum_{i \geqslant 1} \kappa_i^{-1}(g(\lambda))_i^2$$

Since the objective is monotonically increasing in $\kappa_0$, the optimal choice for $\kappa_0$ is the lower bound above $\kappa_0 \geqslant \sum_{i \geqslant 1} \kappa_i^{-1}(g(\lambda))_i^2$. Given this choice, the objective in terms of $\kappa_{1:n}$ becomes

$$\sum_{i \geqslant 1} \kappa_i + \frac{(g(\lambda))_i^2}{\kappa_i}$$

By the AM-GM inequality, the optimal choice for the remaining terms $\kappa_{1:n}$ is then $\kappa_{1:n} = |g(\lambda)|$.   $\square$

# B Experimental details

## B.1 Verifying Adversarial Robustness: Training and Hyperparameter Details

**Optimization details.** We perform subgradient descent using the Adam [27] update rule for MLP experiments, and RMSProp for CNN experiments. We use an initial learning rate of $1e-3$, which we anneal twice by 10. We use 15K optimization steps for all MLP experiments, 60K for CNN experiments on MNIST, and 150K on CIFAR-10. All experiments run on a single P100 GPU.

**Adaptive learning rates**   For MLP experiments, we use an adaptive learning rate for dual variables associated with the constraint $x_{i+1} \odot \big(x_{i+1} - \mathbb{L}_i(x_i)\big) \leq 0$, as mentioned in Section 5.3. In early experiments for MLP-Adv, we observed very sharp curvature in the dual objective with respect to these variables – the gradient has values on the order $\approx 1e3$ while the solution at convergence has values on the order of $\approx 1e{-}2$. Thus, for all MLP experiments, we decrease learning rates associated with these variables by a $10\times$ factor. While SDP-FO produced meaningful bounds even without this adjustment, we observed that this makes optimization significantly more stable for MLP experiments. This adjustment was not necessary for CNN experiments.

**Training Modes**   We conduct experiments on networks trained in three different modes. **Nor** indicates the network was trained only on unperturbed examples, with the standard cross-entropy loss. **Adv** networks use adversarial training [38]. **Mix** networks average the adversarial and normal losses, with equal weights on each. We find that **Mix** training, while providing a significant improvement in test-accuracy, renders the model less verifiable (across verification methods) than training only with adversarial examples.

The suffix **-4** in the network name (e.g. CNN-A-Mix-4) indicates networks trained with the large perturbation radius $\epsilon_{\text{train}} = 4.4/255$. We find that using larger $\epsilon_{\text{train}}$ implicitly facilitates verification at smaller $\epsilon$ (across verification methods), but is accompanied by a significant drop in clean accuracy. For all other networks, we choose $\epsilon_{\text{train}}$ to match the evaluation $\epsilon$: i.e. generally $\epsilon = 0.1$ on MNIST and $\epsilon = 2.2/255$ on CIFAR-10 (which slightly improves adversarial robustness relative to $\epsilon = 2/255$ as reported in [22]).

**Pre-trained networks**   For the networks **MLP-LP**, **MLP-SDP**, **MLP-Adv**, we use the trained parameters from [52], and for the networks **MLP-Nor**, **MLP-Adv-B** we use the trained parameters from [55].

**Model Architectures**   Each model architecture is associated with a prefix for the network name. Table 2 summarizes the CNN model architectures. The MLP models are taken directly from [52, 55] and use fully-connected layers with ReLU activations. The number of neurons per layer is as follows: **MLP-Adv** 784-200-100-50-10, **MLP-LP/MLP-SDP** 784-500-10, **MLP-B/MLP-Nor** 784-100-100-10.

| Model | CNN-A | CNN-B |
|---|---|---|
| | CONV 16 4×4+2 | CONV 32 5×5+2 |
| **Architecture** | CONV 32 4×4+1 | CONV 128 4×4+2 |
| | FC 100 | FC 250 |
| | FC 10 | FC 10 |

Table 2: Architecture of CNN models used on MNIST and CIFAR-10. Each layer (except the last fully connected layer) is followed by ReLU activations. CONV T W×H+S corresponds to a convolutional layer with T filters of size W×H with stride of S in both dimensions. FC T corresponds to a fully connected layer with T output neurons.

## B.2   Verifying VAEs

**Architecture Details**   We train a VAE on the MNIST dataset with the architecture detailed in Table 3.

| Encoder | Decoder |
|---|---|
| FC 512 | FC 1568 |
| FC 512 | CONV-T 32 3×3+2 |
| FC 512 | CONV-T 3×3+1 |
| FC 16 | |

Table 3: The VAE consists of an encoder and a decoder, and the architecture details for both the encoder and the decoder are provided here. CONV-T T W×H+S corresponds to a transpose convolutional layer with T filters of size W×H with stride of S in both dimensions.

(a) Original and Perturbed Digit '9'　　　(b) Original and Perturbed Digit '0'

Figure 3: Two digits from the MNIST data set, and the corresponding images when perturbed with Gaussian noise, whose squared $\ell_2$-norm is equal to the threshold ($\tau = 40.97$). $\tau$ corresponds to threshold on the reconstruction error used in equation (7).

**Optimization details.** We perform subgradient descent using RMSProp with an initial learning rate of $1e-3$, which we anneal twice by $10$. All experiments run on a single P100 GPU, and each verification instance takes under 7 hours to run.

**Computing bounds on the reconstruction loss based on interval bound propagation** Interval bound propagation lets us compute bounds on the activations of the decoder, given bounded $l_\infty$ perturbations in the latent space of the VAE. Given a lower bound $lb$ and an upper bound $ub$ on the output of the decoder, we can compute an upper bound on the reconstruction error $\|s - \hat{s}\|_2$ over all valid latent perturbations as $\|\max\{|ub-s|,|s-lb|\}\|_2^2$, where $\max$ represents the element-wise maximum between the two vectors. We visualize images perturbed by noise corresponding to the threshold $\tau$ on the reconstruction error in Section 6.2 in Figure 3.

## C  Additional results

### C.1  Detailed comparison to off-the-shelf solver

**Setup** We isolate the impact of optimization by comparing performance to an off-the-shelf solver with the same SDP relaxation. For this experiment, we use the MLP-Adv network from [51], selecting quadratic constraints to attain an equivalent relaxation to [51]. We compare across 10 random examples, using the target label with the highest loss under a PGD attack, i.e. the target label closest to being misclassified. For each example, we measure $\underline{\Phi}_{\text{PGD}}$, $\overline{\Phi}_{\text{SDP-IP}}$, and $\overline{\Phi}_{\text{SDP-FO}}$, where $\overline{\Phi}$ and $\underline{\Phi}$ are as defined in Section 6.1. Since the interior point method used by MOSEK can solve SDPs exactly for small-scale problems, this allows analyzing looseness incurred due to the relaxation vs. optimization. In particular, $\overline{\Phi}_{\text{SDP-IP}} - \underline{\Phi}_{\text{PGD}}$ is the relaxation gap, plus any suboptimality for PGD, while $\overline{\Phi}_{\text{SDP-IP}} - \overline{\Phi}_{\text{SDP-FO}}$ is the optimization gap due to inexactness in the SDP-FO dual solution.

**Results** We observe that SDP-FO converges to a near-optimal dual solution in all 10 examples. This is shown in Figure 4. Quantitatively, the relaxation gap $\overline{\Phi}_{\text{SDP-IP}} - \underline{\Phi}_{\text{PGD}}$ has a mean of $0.80$ (standard deviation $0.22$) over the 10 examples, while the optimization gap $\overline{\Phi}_{\text{SDP-IP}} - \overline{\Phi}_{\text{SDP-FO}}$ has a mean of $0.10$ (standard deviation $0.07$), roughly $8\times$ smaller. Thus, SDP-FO presents a significantly more scalable approach, while sacrificing little in precision for this network.

*Remark.* While small-scale problems can be solved exactly with second-order interior point methods, these approaches have poor asymptotic scaling factors. In particular, both the SDP primal and dual problems involve matrix variables with number of elements quadratic in the number of network activations $N$. Solving for the KKT stationarity conditions (e.g. via computing the Cholesky decomposition) then requires memory $O(N^4)$. At a high-level, SDP-FO uses a first-order method to save a quadratic factor, and saves another quadratic factor through use of iterative algorithms to avoid materializing the $M(\lambda)$ matrix. SDP-FO achieves $O(Nk)$ memory usage, where $k$ is the number of Lanczos iterations, and in our experiments, we have found $k \ll N$ suffices for Lanczos convergence.

(a) MNIST, MLP-Adv

Figure 4: *Comparison to off-the-shelf solver.* For 10 examples on MNIST, we plot the verified upper bound on $\phi_x$ against the adversarial lower bound (using a single target label for each), comparing SDP-FO to the optimal SDP bound found with SDP-IP (using MOSEK). In all cases, the SDP-FO bound is very close to the SDP-IP bound, demonstrating that SDP-FO converges to a near-optimal dual solution. Note that in many cases, the scatter points for SDP-FO and SDP-IP are directly overlapping due to the small gap.

## C.2 Investigation of relaxation tightness for MLP-Adv

**Setup**   The discussion above in Appendix C.1 suggests that SDP-FO is a sufficiently reliable optimizer so that the main remaining obstacle to tight verification is tight relaxations. In our main experiments, we use simple interval arithmetic [39, 23] for bound propagation, to match the relaxation in [51]. However, by using CROWN [70] for bound propagation, we can achieve a tighter relaxation.

**Results**   Using CROWN bounds in place of interval arithmetic bounds improves the overall verified accuracy from $83.0\%$ to $91.2\%$. This closes most of the gap to the PGD upper bound of $93.4\%$. For this model, while the SDP relaxation still yields meaningful bounds when provided very loose initial bounds, the SDP relaxation still benefits significantly from tighter initial bounds. More broadly, this suggests that SDP-FO provides a reliable optimizer, which combines naturally with development of tighter SDP relaxations.

## C.3 Verifying Adversarial Robustness: Additional Results

Table 4 provides additional results on verifying adversarial robustness for different perturbation radii and training modes. Here, we consider perturbations and training-modes not included in Table 1. We find that across settings, **SDP-FO** outperforms the **LP**-relaxation.

| Dataset | Epsilon | Model | Training Epsilon | Accuracy Nominal | PGD | Verified Accuracy SDP-FO (Ours) | LP |
|---------|---------|-------|------------------|---------|-----|--------------------|-----|
| MNIST | $\epsilon = 0.3$ | CNN-A-Adv | $\epsilon_{\text{train}} = 0.3$ | $98.6\%$ | $80.0\%$ | $43.4\%$ | $0.2\%$ |
| CIFAR-10 | $\epsilon = \frac{2}{255}$ | CNN-A-Adv | $\epsilon_{\text{train}} = \frac{2.2}{255}$ | $68.7\%$ | $53.8\%$ | $39.6\%$ | $5.8\%$ |
| | | CNN-A-Adv-4 | $\epsilon_{\text{train}} = \frac{4.4}{255}$ | $56.4\%$ | $49.4\%$ | $40.0\%$ | $38.9\%$ |
| | $\epsilon = \frac{8}{255}$ | CNN-A-Adv | $\epsilon_{\text{train}} = \frac{8.8}{255}$ | $46.9\%$ | $30.6\%$ | $18.0\%$ | $3.8\%$ |
| | | CNN-A-Mix | $\epsilon_{\text{train}} = \frac{8.8}{255}$ | $56.7\%$ | $26.4\%$ | $9.0\%$ | $0.1\%$ |

Table 4: Comparison of verified accuracy across various networks and perturbation radii. All SDP-FO numbers computed on the first 100 test set examples, and numbers for **LP** on the first 1000 test set examples. The perturbations and training-modes considered here differ from those in Table 1. For all networks, SDP-FO outperforms the **LP**-relaxation baseline.

## C.4 Comparison between Lanczos and exact eigendecomposition

All final numbers we report use the minimum eigenvalue from an exact eigendecomposition (we use the `eigh` routine available in SciPy [62]). However, the exact decomposition is far too expensive to

use during optimization. On all networks we studied, Lanczos provides a reliable surrogate, while using dramatically less computation. For example, for CNN-A-Mix, the average gap between the exact and Lanczos dual bounds – the values of Equation (5) using the true $\lambda_{\min}$ compared to the Lanczos approximation of $\lambda_{\min}$) – is $0.14$ with standard deviation $0.07$. This gap is small compared to the overall gap between the verified upper and adversarial lower bounds, which has mean $0.60$ with standard deviation $0.22$. We observed similarly reliable Lanczos performance across models, for both image classifier and VAE models in Sections 6.1 and 6.2.

At the same time, Lanczos is dramatically faster than the exact eigendecomposition: roughly $0.1$ seconds (using $200$ Lanczos iterations) compared to $5$ minutes. For the VAE model, this gap is even larger: roughly $0.2$ seconds compared to $2$ hours. For even larger models, it may be infeasible to compute the exact eigendecomposition even once. Although unnecessary in our current work, high-confidence approximation bounds for eigenvectors and associated eigenvalues from Lanczos can be applied in such cases [31, 46].

## Footnotes

[3]The factor of $\frac{1}{2}$ is introduced for convenience