[Reviews · NeurIPS 2020]

Review 1

Summary and Contributions: The authors propose a first-order dual SDP algorithm for verifying properties of verification-agnostic networks. The generalization of the SDP formulation for adversarial robustness is presented. And an algorithm based on forward or backward operations on layers of the neural network is proposed to solve it.

Strengths: The theoretical analysis is sufficient, and the experimental results demonstrate it's efficiency.

Weaknesses: There are several upper bounds constraint, it is better to discuss and compare with those methods.

Correctness: Yes, the claims and method are correct. The empirical methodology is correct.

Clarity: Yes, the paper is well written.

Relation to Prior Work: Yes, it is clearly discussed how this work differs from previous contributions.

Reproducibility: Yes

Additional Feedback: The authors' response is satisfied


Review 2

Summary and Contributions: This paper proposes a first-order variant of SDP-based robustness certification for neural networks. The proposed method is efficiently implementable on GPUs and scales to much larger networks compared to prior works and outperformed LP-based methods.

Strengths: - Strong experimental results.

Weaknesses: - Novelty, review of relevant works - Presentation

Correctness: I tried my best to verify the mathematical claims and found no obvious gaps or mistakes.

Clarity: The writing can be improved in the future version, see detailed comments below.

Relation to Prior Work: The comparison to prior works is somewhat incomplete, especially on the first-order SDP solvers.

Reproducibility: Yes

Additional Feedback: Post-rebuttal update: The authors' response is very nice and detailed, probably the best one I've seen in NeurIPS/ICML this year. It does address many of my previous concerns, including the ones regarding contribution/novelty of this paper. I am raising my score so we can reach a consensus. ------------------- This paper proposes a first-order variant of SDP-based robustness certification for neural networks. The proposed method is efficiently implementable on GPUs and scales to much larger networks compared to prior works and outperformed LP-based methods. The main strength of this paper is it strong empirical performance - using the first order SDP solver proposed in the paper, the authors were able to verify much larger networks, and provided the first SDP-based certification for CIFAR-10. (disclaimer: I am not well-versed enough on the state-of-the-art of SDP based certification, so it's possible that I missed some of the existing work here). However, my main conservation on this paper is about its novelty, presentation and review of the prior works. (1) Novelty and review of the prior works: the main improvement of this paper over existing works [33], [15] is that they proposed the first-order SDP solver instead of the off-the-shelf ones, along with some heuristic acceleration tricks (and if I understood correctly, that's the only difference with [33]). However, the usage of the first-order methods in semidefinite programming is not a novel idea at all. From my quick and incomplete literature review, the origin of this idea can at least be dated back to 2002, in R. Monteiro's paper "First-and second-order methods for semidefinite programming" published in Mathematical Programming. There are some other papers on this topic, for example, Primal-dual first-order methods with O(1/epsilon) iteration-complexity for cone programming, Guanghui Lan, Zhaosong Lu, Renato Monteiro, Mathematical Programming, 2011. I would also like to refer to Zaiwen Wen's 2009 PhD thesis in Columbia University, "First Order Methods for Semidefinite Programming" and the bibliography therein. There are also other approaches to fast SDP solving, e.g. Sanjeev Arora and Satyen Kale's STOC 2007 paper "Combinatorial, primal-dual approach to semidefinite programs". Although I am not an expert in this topic (fast SDP solver), it seems like there is a whole research area dedicated to it, but this paper did not cite any of these papers. In the experiment and algorithm design sections, there is also no comparison with existing first-order SDP solvers. In my opinion, this part requires a major revision. (2) Presentation: It is very difficult to understand Section 4 for anyone who haven't read [33] before (and the presentation in [33] is much more accessible). My suggestion is to include a brief summary of [33] before presenting the lagrangeian formulation. And it should be highlighted more about the fact that proposition 1 is just an lagrangian re-formulation of the SDP in [33]. Regarding the experiments: I hope the authors can explain the discrepancy between the verified accuracy for MLP-SDP using your method and [33]? As the authors mentioned (line 174), both methods are optimizing the same objective, but a 5%+ performance gap seems too large for such scenario. (I noticed that the authors mentioned a probable reason which is the use of different test set subsample, but it still seems a bit too large to me.) Minor comments: - line 163, optimval - optimal


Review 3

Summary and Contributions: This paper considers the semidefinite program for verifying adversarial robustness from [33], and proposes an optimization approach for obtaining an approximation for such SDP that reduces the computational complexity significantly and is still a certified upper bound.

Strengths: The improvement in terms of computational complexity is significant. Typically SDPs are quite inefficient. The authors exploit the specific structure of the SDP to further relax it to an optimization problem that they can approach with first order methods. They implement the first order method by using an automatic differentiation tool. In particular this approach reduces the memory dependence on the number of activations from quartic (in [33]'s if an interior point method is used) to linear. The numerical section is comprehensive.

Weaknesses: I don't see major weaknesses in this paper. One small weakness is that they don't show an explicit measurement of how this paper improves upon [33]. For instance reporting the running times in the experiments (or the appendix) could be a way to do so.

Correctness: The methodology seems correct.

Clarity: The paper is clear. An explicit succinct explanation of how verification methods work could be useful. For instance, what points are chosen to evaluate them, and what are the typical \phi functions taken into consideration, but given the space constraints I think it's quite self-contained. There are some typos the authors should fix (see below).

Relation to Prior Work: Relevant previous work is properly cited.

Reproducibility: Yes

Additional Feedback: The paper is very nice. The optimization ideas are straightforward and the application is relevant. I found many typos. Some of them are below but I am sure I didn't catch them all. line 112: is->be line 116 andb line 142: desribe line 145: relexation line 163: optimval line 215: is->are Algorithm 1: reference to equation (9) should be (4)

[Author Response · NeurIPS 2020]

**Common.** We thank the reviewers for their helpful feedback which has strengthened the paper. All reviewers noted the strong empirical results, and thorough comparisons to other neural network verification approaches. We also note we are currently open-sourcing our code, similar to the version shared in the supplementary material.

**R1** Thank you for the feedback. Regarding comparison with other upper bounding techniques: we compare empirically with the LP-based approach from [Salman, 2019], which subsumes several relevant upper bounding methods such as [Zhang et al., 2018, Weng et al., 2018, Singh et al., 2019, Gowal et al., 2018]. Further, our solver is general purpose and we can directly incorporate additional constraints, e.g. those from [Ehlers, 2017]. We will revise to clarify.

**R2** Thanks for your helpful feedback and pointers. We apologize for oversight in omitting citations to relevant optimization work on first-order SDP solvers. We include a paragraph discussing first-order SDP solvers at the bottom of our response.

**Clarification of contributions:** We do not develop a general-purpose SDP solver or a novel reformulation of semidefinite programming approaches to neural network verification - rather our focus is on *applying* well-known techniques in first-order SDP algorithms to semidefinite relaxations of neural network verification and developing and evaluating a practical implementation that can leverage hardware accelerators (GPUs/TPUs). This will be clarified in our revised paper (Sections 1, 5.1, and 7). Our specific contributions are:

1. Sections 5.1 and 5.2 derive an eigenvalue optimization formulation for neural network verification. While the ideas here are themselves not novel – e.g. Section 3 of Helmberg and Rendl [3] is very similar (we will clarify this in the revision) – we are the first to apply them to neural network verification. Specifically, we show how for neural networks, subgradient computations can be expressed through autodiff of standard network layers, leading to an implementation with linear memory, runtime, and hardware accelerator compatibility.

2. Section 5.3 presents various tricks for initialization, regularization and step-size schedules which enable the strong empirical results demonstrated in Section 6.

3. Section 6 demonstrates that these applications allow us to verify verification-agnostic networks which were intractable for all previous neural network verification methods, as noted by the reviewers. Compared to second-order methods, first-order methods can achieve matching bounds for small networks, while also scaling to verification of mid-size networks where second-order methods become prohibitively expensive.

**Comparisons with [33]:** We have a direct comparison on 10 samples: see Appendix C.1 (Figure 4). The bounds achieved by SDP-FO and SDP-IP [33] almost exactly coincide, with the SDP-IP bound slightly tighter on each. This makes us confident that the two methods produce identical bounds, and the differences in Table 1 are due to sampling noise. We'll add this note to the caption.

> **To be added to Section 7** **First-order SDP solvers:** While interior-point methods are theoretically compelling, the demands of large-scale SDPs motivate first-order solvers. Common themes within this literature include smoothing of nonsmooth objectives [7, 4, 2] and spectral bundle or proximal methods [3, 5, 8]. Many primal-dual algorithms [10, 6, 1] exploit computational advantages of operating in the dual – our dual-based approach to verification naturally inherits these advantages. A full survey is beyond scope here, but we refer interested readers to Tu and Wang [9] for an excellent survey.
>
> Our formulation in Section 5.1 closely follows the eigenvalue optimization formulation from Section 3 of Helmberg and Rendl [3]. We show that within within this formulation, subgradients can be computed using autodiff both easily and efficiently – with linear memory, runtime, and efficient GPU/TPU implementations. While in this work, we show that vanilla subgradient methods are sufficient to achieve practical performance, integrating the ideas from the first-order SDP solver literature mentioned above is a promising candidate for future work, and could potentially allow faster convergence in practice.

**R3** Thank you for the feedback. We have fixed the typos and added the paragraph below regarding run-times. Note the main advantage relative to [33] is that the SDP-IP approach is simply intractable for our CNN models due to their size.

> **To be added to Section 6** **Computational resources:** Using a P100 GPU, maximum runtime for our approach is roughly 15 minutes for all MLP instances, and 3 hours for CNN instances, though most instances are verified sooner. For reference, SDP-IP [33] uses 25 minutes on a 4-core CPU for MLP instances, and is intractable for CNN instances due to $O(n^4)$ memory usage.

[1] Sanjeev Arora and Satyen Kale. A combinatorial, primal-dual approach to semidefinite programs. 2016.
[2] Alexandre d'Aspremont and Noureddine El Karoui. A stochastic smoothing algorithm for semidefinite programming. 2014.
[3] Christoph Helmberg and Franz Rendl. A spectral bundle method for semidefinite programming. 2000.
[4] Guanghui Lan, Zhaosong Lu, and Renato DC Monteiro. Primal-dual first-order methods with O($1/\epsilon$)... 2011.
[5] Claude Lemaréchal and François Oustry. Nonsmooth algorithms to solve semidefinite programs. 2000.
[6] Renato DC Monteiro. First-and second-order methods for semidefinite programming. 2003.
[7] Yurii Nesterov. Smoothing technique and its applications in semidefinite optimization. 2007.
[8] Neal Parikh and Stephen Boyd. Proximal algorithms. 2014.
[9] Stephen Tu and Jingyan Wang. Practical first order methods for large scale semidefinite programming. 2014.
[10] Zaiwen Wen. First-order methods for semidefinite programming. 2009.


[Meta-Review · NeurIPS 2020]

There was a consensus among all reviewers on accepting the paper. The paper clarity and the clarity of the rebuttal were appreciated by the reviewers and it addressed all of the concerns. Please include the rebuttal answers in the final version of the paper.